# piNET–An Automated Proliferation Index Calculator Framework for Ki67 Breast Cancer Images

**DOI:** 10.3390/cancers13010011

**Published:** 2020-12-22

**Authors:** Rokshana Stephny Geread, Abishika Sivanandarajah, Emily Rita Brouwer, Geoffrey A. Wood, Dimitrios Androutsos, Hala Faragalla, April Khademi

**Affiliations:** 1Electrical, Computer and Biomedical Engineering Department, Ryerson University, Toronto, ON M5B 2K3, Canada; a.sivanandarajah@gmail.com (A.S.); dimitri@ryerson.ca (D.A.); 2Department of Pathobiology, Ontario Veterinarian College, University of Guelph, Guelph, ON NIG 2W1, Canada; ebrouwer@uoguelph.ca (E.R.B.); gewood@uoguelph.ca (G.A.W.); 3Department of Laboratory Medicine & Pathobiology, St. Michael’s Hospital, Unity Health Network, Toronto, ON M5B 1W8, Canada; Hala.Faragalla@unityhealth.to; 4Keenan Research Center for Biomedical Science, St. Michael’s Hospital, Unity Health Network, Toronto, ON M5B 1W8, Canada

**Keywords:** breast cancer, deep learning, digital pathology, hematoxylin, Ki67, proliferation index

## Abstract

**Simple Summary:**

Approximately 2.1 million women are affected by breast cancer every year. Invasive disease accounts for 80% of breast cancer cases and is the most common and aggressive type of breast cancer. Early diagnosis is the key to survival. Ki67 biomarkers have been shown to be a promising prognostic biomarker in this regard, but manual proliferation index (PI) calculation is time consuming and subject to inter/intra observer variability which reduces clinical utility. Computational pathology tools can aid pathologists to make the diagnostic process more efficient and accurate. With the advent of deep learning, there is great promise that this technology can solve problems that were difficult to tackle in the past, but more work needs to be done to combat the challenge of multi-center datasets. In this work, a novel Ki67 PI calculator based on deep learning is proposed, called piNET, which is shown to be accurate, reliable, and consistent across multi-center datasets.

**Abstract:**

In this work, a novel proliferation index (PI) calculator for Ki67 images called piNET is proposed. It is successfully tested on four datasets, from three scanners comprised of patches, tissue microarrays (TMAs) and whole slide images (WSI), representing a diverse multi-centre dataset for evaluating Ki67 quantification. Compared to state-of-the-art methods, piNET consistently performs the best over all datasets with an average PI difference of 5.603%, PI accuracy rate of 86% and correlation coefficient R = 0.927. The success of the system can be attributed to several innovations. Firstly, this tool is built based on deep learning, which can adapt to wide variability of medical images—and it was posed as a detection problem to mimic pathologists’ workflow which improves accuracy and efficiency. Secondly, the system is trained purely on tumor cells, which reduces false positives from non-tumor cells without needing the usual pre-requisite tumor segmentation step for Ki67 quantification. Thirdly, the concept of learning background regions through weak supervision is introduced, by providing the system with ideal and non-ideal (artifact) patches that further reduces false positives. Lastly, a novel hotspot analysis is proposed to allow automated methods to score patches from WSI that contain “significant” activity.

## 1. Introduction

Breast cancer is the second most frequently diagnosed cancer worldwide and the leading cause of cancer-related deaths in women [1]. Approximately 2.1 million women every year are impacted, in 2018, approximately 627,000 women died from the disease. Invasive ductal carcinoma (IDC) accounts for 80% of breast cancer cases and is the most invasive of the breast cancers. Over time, if left untreated, IDC can metastasize to other areas of the body [2]. Breast cancer survival is attributed largely to timely and accurate diagnosis, which is dependent upon many features including tumor size, tumor cell proliferation and morphological features of the tumor [3,4]. Histopathology plays a vital role in obtaining these measurements, and is used for diagnosing breast cancer disease, patient management and for predicting prognosis [3]. Traditionally, pathologists examine tissue under magnification with hematoxylin and eosin (H&E) stains to highlight cellular morphology and tissue microstructure for diagnosis and tumor grading. Grading systems such as Bloom-Richardson [5] are used to report the mitotic index, as well as the nuclear and tubule grades for breast tumors.

To characterize tumors further and to increase response to therapy via targeted approaches, there is a growing interest in using immunohistochemical (IHC) biomarkers [6]. A promising IHC biomarker for breast cancer is MKi67 (or Ki67), a protein that is indicative of cell proliferation and it is found in all active phases of the mitotic cell cycle [7]. Ki67 tumor proliferation rates have been clinically shown to be correlated with tumor aggressiveness, predicted survival rates, reoccurrence and also has great potential in improving decisions around treatment options [6,7,8,9]. Although integrating Ki67 into clinical workflows could improve quality of care, computing Ki67 proliferation indices (PI) is time consuming and subject to inter- and intra-observer variability among pathologists [10]. The American Society of Clinical Oncology and European Society for Medical Oncology (ESMO) have stated that Ki67 would be a useful clinical tool if it were standardized [9,11]. Moreover, the number of pathologists in the workforce is declining, and the number of cancer cases is increasing [12,13] which creates pressures on pathologists. They are expected to complete more cases in less time. Pathologists working more than 39 h per week are found to be fatigued, overworked, and burned out [12,13,14,15], which can affect quality of care. Therefore, to increase clinical applicability and overall utility of Ki67 biomarkers, PI measurements should be standardized and efficient. Digital pathology, a relatively new technology, can be leveraged to address these challenges.

Technological advances in whole slide scanners are creating high resolution digital images that can be analyzed automatically with image analysis, machine learning and artificial intelligence (AI). In fact, digital pathology is expected to be the “first frontier” of medical AI [16]. Computational pathology methods could standardize PI intervals and be used to study PI in large cohorts efficiently. Scoring tools that provide automated PI measures that would improve workflow efficiency, turn-around-times, reduce subjectivity and ultimately improving patient care. With the emergence of deep learning techniques achieving high performance in segmentation and classification tasks for images, there is potential for these methods to revolutionize the way digital pathology images are analyzed.

Automated PI quantification is an on-going research topic due to the challenging nature of the task. There is variability in stain and scanner vendors which contributes to an overall lack of reproducibility [17]. In [18] we proposed an IHC color histogram (IHCCH) approach to automatically compute the PI for TMAs in an unsupervised manner which achieved 92.5% PI measurement accuracy on two datasets (80 TMAs). This method was developed to detect any cells that was stained with Diaminobenzidine (DAB or Ki67+) or Hematoxylin (Ki67−). Since the TMAs largely contain tumor, there was high PI estimation performance. However, there are challenges when this method is when applied to new clinical datasets that contain both tumor and non-tumor cells which are both detected and reduces PI estimation performance. Additionally, different types of tissues, cells and artifact can all disrupt the overall PI measurement by mainly creating false positives in the nuclei detection phase.

The accurate detection of tumor cells is a key factor in accurate calculation of PI. The importance of distinguishing between cell morphologies was explored in [19] where two commercially available software tools were used to quantify PI in 53 breast cancer WSI biopsies from a single center. It was found that cases with PI differences of ≥10% were caused by errors in tumor cell classification, overlapping cells or staining artifacts.

To overcome the inclusion of non-tumor cells and false positives in PI quantification, several algorithms have been proposed to address these challenges. In [20], PTM-NET was introduced to analyze Ki67 stained breast cancer whole slide images with machine learning to segment tumor regions which are then supplied to a PI calculator. The PTM-NET achieved a dice correlation coefficient of 0.74 between the automated and manually delineated tumor regions from 87 patients acquired across two centers. To try and reduce the impact of non-tumor regions, in [21], the authors use a U-NET model to segment unwanted regions such as folds, smears and color distortions due to staining variability in Ki67 stained brain tumor specimens. If this work could translate to breast cancer tissue, these regions could potentially be used to remove false positives in the nuclei detection phase to increase PI calculation accuracy. In [22], Xing et al. propose, KiNET, a deep learning method is used to detect nuclei, perform tumor region segmentations, and classify between tumor and nontumor nuclei in one framework. The model was evaluated on 38 Ki67 pancreatic NET cases of size 500 × 500 and achieved an F1-score of 0.898 and 0.804 for nuclei detection and classification respectively. Although promising, the method requires several annotations and ground truth types, which can be cumbersome to obtain. In [23], a three stage algorithm was proposed for Ki67 scoring, which included various algorithms to segment cell boundaries, extract nuclei features and classify each cell as tumor or non-tumor, which was then used to compute the PI. The challenge is that individual nuclei boundaries were segmented–these types of ground truths are time consuming to generate and there can be ambiguity in defining cell boundaries in overlapping nuclei.

Deep learning has become a prominent choice for analyzing and quantifying digital pathology images. In [24], over 130 papers using various datasets and machine learning strategies for digital pathology, including fully supervised, weakly supervised, unsupervised, transfer learning and various variants that were critically analyzed. Despite the excitement of using AI in computational pathology, a single unified approach does not exist. There are many challenges that remain which should be addressed for effective and reliable PI calculation in one framework. For example, other works have tried to mitigate the inclusion of non-tumor cells through tumor preprocessing or cell classification. These types of methods can be laborious to develop (large amounts of annotations required) and any error in this phase is propagated down to the PI computation step. Additionally, image artifacts, cell clustering, folds and non-tumor cells all create false positives and although [21] work shows promise by detecting artifact regions, this work has not been translated to breast tissue or been used to specifically aid in nuclei detection. Moreover, many of these works are tested in single center data with limited variability. Therefore, it is difficult to determine if the methods would generalize to new, unseen data.

In this work, we present piNET, a novel U-NET [25]-based model for PI quantification of Ki67 breast cancer tissue to overcome the challenges of previous approaches. It automatically detects immuno-positive and immuno-negative tumor nuclei separately within a single framework, and suppresses false positives using novel sampling strategies. Therefore, accurate tumor preprocessing is not required, and false positives reduction schemes are not needed to reduce issues created by folds, artifacts and nontumor cells/tissues. Performance is evaluated in 773 regions of interest (ROI), 86 TMAs and 55 WSIs (three image types), from four institutions, three scanner models and two types of mammary tissue (human and canine). It comprises of one of the largest multi-centre Ki67 datasets evaluated in the literature.

The framework is trained and tested on individually annotated tumor nuclei, to ensure stromal, epithelial, and other non-tumor cell types, are not included in the final PI calculation. This offers an accurate PI measurement without dependence on tumor preprocessing. We strictly focus on nuclei detection (instead of segmentation) to mimic a pathologist’s workflow in performing PI quantification, and to simplify ground truth generation. This ground truth generation approach is faster and is more accurate to annotate nuclei centers versus boundaries. Based on the nuclei center annotations, a Gaussian kernel is generated, and different channels are used for the Ki67+ and Ki67− nuclei so that both tumor nuclei types are annotated in a single RGB image. Motivation for using a Gaussian kernel is to allow the system to contextual learn information from the nuclei (versus a single point in the middle), which could help the classifier discover more robust features. These ground truths are then used to train and test the U-NET architecture, which detects Ki67+/− tumor nuclei. Regression-based loss functions are utilized used since the ground truth and predictions are non-binary. A novel data sampling strategy is introduced which includes the concept of ideal and non-ideal patches. Ideal patches have mainly tumor nuclei and no artifacts. Non-ideal patches can consist of tumor nuclei, but also consist of folds, artifacts, overstained regions and non-tumor cells/tissues to allow the classifier to learn these regions as part of the background class. Various fine-tuning is completed to find the optimal loss function and data splits. The model is trained on a single dataset and tested on large and diverse datasets for Ki67 which includes patches (ROI), TMAs and WSI data. Evaluation on this type of multi-center data will help to determine the generalizability and overall robustness of the proposed piNET for clinical use.

## 2. Materials and Methods

In this section, the data and methods used to implement the piNET will be outlined. The experimental datasets are from five datasets from four different centers, and the images range from regions of interest (ROIs), tissue micro-arrays (TMAs) and whole slide images (WSIs). Three datasets have ground truth annotations for individual nuclei and the other two have ground truth PI indices or ranges. Of the five datasets, four are of human tissue and one is from canine mammary tissue. The proposed piNET architecture is shown in Figure 1 and includes data preparation, where if the image is a TMA or WSI it is tiled, followed by automated tumor nuclei detection using deep learning. The detected nuclei from piNET, are used for PI computation.

### 2.1. Datasets

Table 1 summarizes the Ki67 stained breast cancer datasets used, along with the scanner type, magnification, ground truth available and number of images. In total, there are five multi-institutional datasets resulting in a total of 773 patches (ROI), 55 WSI and 90 TMAs. SMH patches were 256 × 256 in size and DeepSlides were 512 × 512 (which were down sampled to ×20 yielding patches of 256 × 256). Data is from St. Michael’s Hospital “SMH” in Toronto, Canada, which includes breast biopsies (WSIs) and corresponding patched regions of interest (ROI), ROI patches from an open source dataset “DeepSlides” [26], TMAs from Protein Atlas (open source) [27], and TMAs from the Ontario Veterinary College at the University of Guelph “OVC”. This diverse dataset represents one of the largest multi-center Ki67 datasets analyzed in the literature. It consists of WSIs, patched images and TMAs, as well as images acquired by different scanning devices, from varying pathology laboratories, and a variety of staining variations. Throughout the paper, the model was trained, validated and tested on SMH patches while the other datasets were held out and used to test the pipelines accuracy and generalizability to different image sources and types-ROIs, WSIs and TMAs. These datasets will be used to test the proposed piNET system’s PI quantification accuracy, robustness, and generalizability. Visual representation of each dataset with the corresponding ground truth can be seen in Figure 2.

SMH breast tumor biopsies were stained with DAB (Ki67) and hematoxylin for 55 patients and from each patient a single representative slide was digitized at 20× using Aperio AT Turbo scanner. The SMH WSIs were further patched (described below) to generate annotations for Ki67+/− nuclei. Deepslides [26] contained Ki67 regions of interest (1024 × 1024) which were cropped from 32 different breast cancer patients’ WSIs, scanned using the Aperio ScanScope at 40× magnification with a pixel size of 0.2461 × 0.2461 μm^2^. A total of 113 randomly selected 512 × 512 tiles were cropped and annotated, followed by down sampling to 256 × 256 (to bring the effective resolution down to 20× in order to match the training data). The Protein Atlas [27] dataset consists of 56 Ki67 TMAs from the IHC lab in Uppsala, Sweden, which were acquired by Aperio ScanScope AT and Aperio ScanScope T2 scanners, using a magnification of ×20. The last dataset used consisted of 30 canine breast cancer TMAs stained for Ki67 with 129,404 individually labeled nuclei from the Ontario Veterinary College (OVC) at the University of Guelph. OVC data was used to test the robustness of the pipeline to different mammary cell types, but of same disease. The dataset was scanned using a Leica SCN400 slide scanner, with ×20 magnification and were stained using Ki67 and hematoxylin. The total dataset consists of four databases, three image types (ROIs, WSI and TMAs), three scanner types and three different laboratory staining protocols–representing a diverse set of test images.

SMH, DeepSlide and OVC all have individual labels for both the immuno-positive and immuno-negative tumor cells and annotations were completed by marking the center using ImageJ, with different color markers for positive (Ki67+) and negative (Ki67−) cells [28]. The patched data (SMH and DeepSlide) were annotated by an experienced biomedical student, who was trained by a breast pathologist and only tumor cells were annotated. The OVC TMAs were annotated by a DVM from the OVC department of pathobiology and the cores contained mainly tumorous tissue. Ground truth PI of these three datasets were calculated using the expert nuclei annotations. To generate SMH patches, SMH biopsies (WSIs) were cropped in Pathcore’s Sedeen viewer [29]. Only regions that held representative information of the overall slide were considered. SMH biopsies were scored for PI by an experienced breast pathologist. The pathologist initially scanned the entire whole slide image to distinguish between homogenous, heterogenous or hotspot pattern in proliferation. Afterwards, three representative regions across the entire tumor were identified to reflect the average proliferation across the tumor and total of 500 cells were included in the proliferation calculation. These representative regions were not disclosed. There were 55 corresponding Phosphohistone H3 (PHH3) stained slides with the manually found mitotic index. Protein Atlas supplied PI ranges (i.e., Low: PI < 10%, Medium 25% ≤ PI ≤ 75%, High: PI > 75%) as opposed to individually annotated nuclei or specific PI scores.

When cropping ROIs from SMH WSIs, 330 patches of ideal regions along with 330 patches of nonideal regions were cropped over the entire dataset. Ideal patches are defined as regions where ‘pristine’ data can be visualized, mainly consisting of only tumorous tissue (i.e., immuno-positive and tumor immuno-negative cells), with minimal nontumor cells/tissue. Nonideal patches may contain tumor cells, but they also include folds, artifacts, nontumor cells, including epithelial cells, inflammatory cells, and excessive tissue, such as stromal or adipose tissue. The importance of distinguishing between ideal and nonideal patches of data will be thoroughly discussed later.

In this work, a single dataset (SMH patches) is used for model tuning, training and validation purposes. All four datasets will be used for testing piNET, which includes SMH patches and WSIs, as well as the other datasets not seen by the classifier to examine generalizability: DeepSlides, Protein Atlas and OVC. We use the SMH patched data to train and validate the proposed piNET system since SMH data is acquired natively at 20× (which aligns with the majority of datasets used), it consists of individual nuclei annotations and high quality images, which is ideal when training a model. Individually annotated nuclei ground truths will be used to examine nuclei detection performance of piNET using F1-Scores and PI values will be used to assess accuracy and reproducibility. To determine PI accuracy all datasets are used: computed PI from the patched data (SMH, DeepSlides, OVC), PI ranges from the Protein Atlas TMAs, and whole slide (patient-level) PI from SMH WSIs.

### 2.2. Methods

In this study, we propose a novel deep learning-based pipeline called piNET which detects tumor cells and differentiates between immunopositive (Ki67+ or DAB) and immunonegative (Ki67− or hematoxylin) cells in one framework. The piNET is constructed from a multiclass U-NET [25] based system trained on tumor cells from SMH patches, and regression losses are used [30]. In images with individually annotated nuclei, a Gaussian proximity map is used to demarcate the nuclei. Based on detected Ki67+/− tumor cells, a proliferation index (PI) is computed for any type of image (TMA, ROI or WSI). As will be shown, one of the major contributions is the architecture’s overall robustness is that the model can quantify Ki67 PI for a variety of image types (ROI, TMAs, WSIs), scanner/stain vendors, lab staining protocols and the presence of artifacts or non-tumor cells. This is investigated using different dataset sources, image types, magnification levels and scanner types. The different datasets were evaluated throughout using various validation metrics depending on the ground truth type, which will be explored in the Validation Metrics section. The overall pipeline is shown in Figure 1 and consists of three major components, (i) Patch Extraction, (ii) piNet: Ki67+/− Detection and (iii) Proliferation Index Quantification. These major methods will be discussed throughout the paper.

#### 2.2.1. Dataset Preparation

In this proposed framework, we investigate a fully automatic pipeline, which quantifies Ki67 PI of breast cancer tissue for various image types, including TMAs, WSIs and patch-based analysis. The piNET architecture has been designed to accept 256 × 256 images, therefore when evaluating TMAs and WSI the images must be preprocessed and tiled first. TMAs are directly tiled into patches of 256 × 256 since they contain mostly relevant tissue to be analyzed for PI quantification. On the other hand, WSI tumor resections have large areas of normal regions, artifacts, fat etc. Therefore, to efficiently speed up processing (number of tiles to analyze) and to isolate regions of interest, the WSI undergoes thresholding (described next) to find candidate analysis regions, i.e., Figure 3, which are then patched into tiles.

Figure 4 illustrates the three steps that are performed to segment candidate areas of interest from the WSI and reduce the number of patches to process. First, the lowest resolution level of the Aperio ScanScope Virtual Slide (svs) file is extracted. WSIs are stored in a pyramid-based structures, where each level of the pyramid has a different level of resolutions and dimensions. Next, Otsu thresholding is applied on lowest resolution version of the image’s red channel [31,32] to detect regions of interest. Lastly, the thresholded segmentation mask is used to tile the WSI. To obtain the tiles in the original resolution, the coordinates from the lowest resolution are upscaled and used to extract the image. The region is then tiled and inputted into piNET for nuclei detection and PI quantification.

#### 2.2.2. piNET: Automated Ki67+/− Detection of Tumor Cells

Automatically detecting and differentiating between positive and negative tumor cells on multicenter datasets is a challenging task and is the central design issue for automated PI computation [33,34]. Different laboratory practices as well as varying staining vendors, scanner manufactures, staining protocols or biomarker expression levels can reduce accuracy of the approaches. Variation within digital pathology imaging data can cause robustness issues within algorithms or lead to overfitting of a certain dataset. Moreover, there are many non-tumor cell types that can interfere with automated PI computation which traditionally required an accurate tumor segmentation preprocessing step [35,36,37]. In this subsection, we introduce piNET; a U-NET-based deep learning architecture that overcomes these challenges. It is a multiclass framework where Ki67+/− tumor nuclei are detected at the same time across multicenter, clinical images. Since tumor nuclei are detected, tumor preprocessing is not required in tiled images. The success of piNET is due to several factors. Firstly, due to the nature of deep learning, the method is able to adapt to variability in images much better than traditional approaches. Secondly, the proposed work detects Ki67+ and Ki67− tumor cells in a single framework, which mimics the pathologist’s workflow, and differentiates between normal and tumor cells for accurate PI quantification. As will be shown piNET can be used to automatically compute PI given patches from TMAs or WSI, which allows the tool to be applied on a variety of realistic, clinical data. Lastly, the model is tuned to be robust to artifacts, folds or noise within the image using a novel data sampling strategy.

The piNET pipeline was built based on the original U-NET architecture, a convolutional neural network (CNN) with an encoding arm which compresses the images into a compact representation, and a decoding arm that reconstructs the prediction. Figure 5 illustrates the architecture used in this work. The original U-NET architecture was designed for segmenting biomedical images with excellent results in variable biomedical data [25], where the model used a per-pixel classification approach to segment cells, followed by watershed to separate overlapping cells in the prediction. Instead, in this method, we use a Gaussian defined proximity map as ground truths for individual nuclei and a regression-based loss function on a per-pixel basis to identify central regions of the tumor nuclei. The model contains six levels of depth, with an initial number of filters set to 32, all the way down to the last layer which ends at 1024. The final activation layer used is the rectified linear unit (ReLU) as ReLU-based models are scale-invariant, are faster than sigmoid, have better gradient propagation and are computationally efficient [30,38,39]. To ensure the model can robustly detect Ki67+ (immuno-positive) or Ki67− (immuno-negative) and omit nontumor cells/tissues/artifacts three classes are used: Ki67+ (red channel), Ki67− (green channel) and non-tumor cells/tissues are comprised of the background. The remainder design aspects of the method are described next.

#### 2.2.3. Ground Truth Generation

For PI quantification, pathologists do not consider the boundaries of the nuclei, but instead are interested in a raw nuclei count of immunopositive (Ki67+) and immunonegative (Ki67−) tumor cells. Therefore, in this work, instead of segmentation, we are focused on automatic detection of immuno-positive and immuno-negative nuclei to mimic pathologists’ workflow. The proposed method also allows for faster, efficient ground truth generation, as tumor cell nuclei centers are annotated with a single seed, rather than segmenting cell boundaries. This method of using detection markers for annotating tumor cells mimics the pathologist’s workflow and is also gaining traction, as more deep learning algorithms are beginning to use this highly efficient method [22,37,38]. Although convenient and in line with pathological analysis, single pixel point annotations for the Ki67+/− nuclei is not enough information to train a deep learning system. Instead, a Gaussian kernel is generated and placed in the center of each nucleus, based on the expert annotation. Given an image $I\left(x_{1},x_{2}\right)$ with $\left(x_{1},x_{2}\right)\;\in \;Z^{2}$, and a nuclei center at the pixel spatial location $\left(x_{1},x_{2}\right)$, a Gaussian kernel is generated:(1)$$g\left(x_{1},x_{2}\right)=\frac{1}{2\pi \sigma^{2}}\;e{\;}^{\frac{-x_{1}^{2}\;+\;x_{2}^{2}}{2\sigma^{2}}}$$
where σ is the standard deviation (or the width of the function). The Gaussian kernel emphasizes the center pixel (highest weight) and gives increasingly less importance to pixels further away from the center. This representation can be beneficial since it incorporates spatial context from within the cell, but places highest priority on the center. This extra contextual information may be beneficial to the CNN layers, since additional information from within the cells can be considered. We call this a spatial proximity map. A small kernel was used: σ=3 pixels, to ensure that the Gaussian does not extend past the boundary of the cell. Since all images are 20× this kernel represents approximately the same size across the datasets. The individual spatial proximity maps of tumor cell centers (for Ki67+/− cells) are shown in Figure 6. Please note that for visualization purposes, Ki67+ tumor cells are red, whereas the Ki67− cells are green, but when training the system each class comprises its own channel.

After obtaining the piNET’s RGB prediction map, the image is processed per marker stain type and the red channel corresponds to predicted Ki67+ tumor cells and the green channel for Ki67− tumor cells. Each map is thresholded using the Otsu [31] method and the count of the number of Ki67+ and Ki67− tumor cells is obtained to quantify the PI.

#### 2.2.4. Loss Functions

A loss function is a critical design element in deep learning since it determines the way weights are updated [30]. Given a prediction from the system and the corresponding ground truth, the goal of the loss function is to measure the similarity between them and to drive the system to learn parameters that maximize this similarity through backpropagation. Loss functions can be categorized as regression- or classification-type. Classification loss functions focus on solving a categorization issue, where the predicted value is binary or categorical [30,40,41]. Segmentation problems are classification problems on a per-pixel basis. Regression-based problems use losses that can predict a quantity (versus a binary or categorical response).

For the detection of Ki67+/− nuclei, we pose this as a regression problem, since the Gaussian kernels are non-binary and the prediction should represent the proximity to the center of the nuclei. Recall from Figure 6 that each tumor cell annotation has a Gaussian distribution with the highest value close to the center of the nuclei. For piNET, we have separate markers for the Ki67+/− nuclei and regress both separately and within one framework. The gradient of the loss function is used to update the weights at the same time for both of the classes separately.

The selection of a loss function is critical for the performance and accuracy of a model [30]. Hence, several regression loss functions are investigated: Huber, LogCosh, Mean Squared Error (MSE) and Root Mean Squared Error (RMSE). Given a predicted output image $\hat{y}\left(x_{1},x_{2}\right)$, the corresponding ground truth $y\left(x_{1},x_{2}\right)$, both with n samples, these loss functions are shown in Table 2. The Huber loss is a robust loss function which shows great potential for regression tasks [42]. A drawback is the manual selection of delta (∂) and for this work, the default ∂=0.1 was used. Huber loss is known to be less sensitive to outliers, in comparison to other loss functions [42,43,44]. LogCosh is another commonly used regression loss function based on the logarithm of the hyperbolic cosine of the difference between prediction and expected. LogCosh may be able to suppress outlier data to result in more robust optimization [44,45]. Mean squared error (MSE) is another intuitive loss function, which is the squared difference between actual and predicted output across the entire training dataset. A potential drawback of MSE is if outliers are present, it can cause heavily penalized values from the squaring operation for outliers. The last loss function to be analyzed is Root Mean Squared Error (RMSE); the MSE square-rooted which is less sensitive to outliers [30,40,43]. As generalizability and adaptability are considered for the design of piNet, the model must be able to accurately detect tumor cells regardless of the presence of folds, artifacts or other cell and tissue types. All four loss functions are investigated to determine which is best for these tasks.

#### 2.2.5. Data Sampling

In Ki67 breast cancer imaging data, there are artifacts, folds, and normal tissues/cells which traditionally create false positives for nuclei detection [19,22]. This can inflate estimates of tumor nuclei counts which negatively affects PI accuracy. Therefore, as part of the design, we considered a few novel data sampling strategies to help piNET learn the representations of Ki67+/− tumor nuclei while suppressing irrelevant background regions, in attempt to reduce false positives. We considered two types of patch types: ideal and non-ideal. Ideal patches are those that are “pristine” in some sense; they have mostly tumor cells and no artifacts or folds. Non-ideal patches on the other hand may have tumor cells, but also consists of folds, artifacts, and other types of non-tumor cells. These non-ideal patches may help the model generalize by allowing the model to learn non-tumor cells and artifacts as part of the background class. The hypothesis is that perhaps data partitions with more non-ideal samples will weakly supervise learning in order to help reduce false positives caused by folds, artifacts and other non-tumor cells since these regions are learned as part of the background class. Annotations follow the same protocol for both the ideal and non-ideal patches, where tumor nuclei are annotated in the center, for two classes: Ki67+ and Ki67− nuclei. Examples of ideal and non-ideal patches are shown in Figure 7.

Three data partitions were investigated throughout this experiment. First, the 100–0 split which is 100% ideal patches and 0 non-ideal patches. The second split is 50–50 where 50% ideal and 50% non-ideal patches of data are used. The last split is 70–30 to give more importance to the pristine data (70% ideal), with still some importance given to the non-ideal scenarios. For each datasplit, we use a 70:10:20 training, testing and validation strategy stratified on PI ranges (low, medium, high).

#### 2.2.6. Proliferation Index Quantification

The output of piNET is a multi-channel prediction image, with center locations of tumor nuclei detected for two classes of Ki67+ and Ki67− cells, and the third channel is the background class. Otsu’s thresholding is used to convert the predictions into binary representations and watershedding is applied to separate any possible overlapping predictions. The number of Ki67+ and Ki67− cells can be easily counted from the binary maps and used to compute the proliferation index (PI):(2)$$\mathrm{Proliferation}\;\mathrm{Index}\;\left(\mathrm{PI}\right)=\;\frac{\;\#\;\mathrm{of}\;\mathrm{Ki}67^{+}\;\mathrm{Tumor}\;\mathrm{Cells}\;}{\mathrm{Total}\;\#\;\mathrm{of}\;\mathrm{Ki}67^{+}+\;\mathrm{Ki}67^{-}\;\mathrm{Tumor}\;\mathrm{Cells}\;}$$

For a given region of interest or patch, Equation (2) gives the effective PI for that region. However, TMAs and WSI are comprised of many patches, therefore there needs to be a way to compute the PI from the recomposed images (i.e., all patches). In TMAs we recompose the images and compute the total PI for the TMAs using Equation (2). On the other hand, if a WSI is being analyzed, a pathologist will typically select hotspots or representative regions for analysis. To mimic this analysis and ideally use similar hotspot regions we proposal a novel method to perform WSI PI quantification. In particular, we find the average number of tumor nuclei per patch denoted $N_{avg}$ and use this as a threshold to find patches with “significant activity” that should be included in the final PI calculation. Namely, only patches with more tumor nuclei than $N_{avg}$ will be included in the final PI. Regions that adhere to this have the total number of Ki67+ and Ki67− nuclei (separately) accumulated over all patches. Given *N(i)* number of tumor cells for patch number *i*, this can be represented in the following way
(3)$$\mathrm{Ki}67^{+\;}\;=\left\{\overset{k}{\underset{i=1}{\sum }}\#\;\mathrm{of}\;\mathrm{Ki}67\;\mathrm{Positive}\;\mathrm{Tumor}\;\mathrm{Cells},\;if\;N\left(i\right)\;\geq \;N_{avg}\;\right\}\;$$
(4)$$\mathrm{Ki}67^{-}=\left\{\overset{k}{\underset{i=1}{\sum }}\#\;\mathrm{of}\;\mathrm{Ki}67\;\mathrm{Negative}\;\mathrm{Tumor}\;\mathrm{Cells},\;if\;N\left(i\right)\;\geq \;N_{avg}\right\}$$
where k represents the number of patches within the WSI. The total accumulated Ki67+/− values, for active regions, are then used to calculate PI:(5)$$\mathrm{WSI}\;\mathrm{PI}=\;\frac{\mathrm{Ki}67^{+}\;}{\;\mathrm{Ki}67^{+}\;+\;\mathrm{Ki}67^{-}}$$

## 3. Results

Evaluation of the proposed piNET algorithm for nuclei detection and PI computation across five multi-centre datasets, including patched regions, TMAs and WSI, will be presented here. Firstly, the piNET model is tuned, and the effect of the design parameters for nuclei detection are evaluated for the proposed (i) Loss functions and (ii) Data Partitions. An optimal design is selected, and the final model is used for further testing of generalizability to unseen data. Secondly, the performance is then analyzed as a function of nuclei detection performance and “Accuracy” for both Ki67+ and Ki67− tumor nuclei using SMH as the test set. To examine “Generalizability”, we then explore the performance of the architecture on two unseen datasets (DeepSlides and OVC) as a function of nuclei detection and PI accuracy rate using Low (PI < 10), Medium (10–30) and High (PI > 30). As previously discussed, DeepSlides consists of human tissues, while the OVC dataset is mammary data. Although, the canine species may express genetic and physiologic similarities to humans [46], the microstructure of the tissues may vary. The model’s ability to generalize while accurately analyzing the contrasting tissues microstructure investigated. Finally, the piNET is compared to three deep learning methods, Res-UNet [47], Dense UNet [48], FCN8 [49], and the previous unsupervised approach, IHCCH [18], across five datasets.

All piNET models are trained with SMH patches, with 100 epochs, batch size of 16, learning rate of 0.001 and Adam optimizer. The model was trained using stratified data of PI ranges (Low PI < 10, Medium PI 10–30 and High PI > 30), of 70:10:20 training, testing and validation (SMH patches). The input image is the 256 × 256 × 3 RGB pathology image with the corresponding ground truth represented as another 256 × 256 × 3 image with three classes, Ki67+ (green channel), Ki67− (red channel), background (black), see Figure 6. The final prediction map is thresholded using Otsu [31] to highlight the most prominent predictions, and to find the corresponding nuclei centers maxima locations. If the cells detected are overlapped, a watershed method is used for separation. The binary detection result is used to calculate performance of the system as a function of F1 Scores, accuracy rates, PI difference and Pearson related coefficients.

### 3.1. Validation Metrics

To validate the accuracy of piNET, a series of validation metrics were considered that measured nuclei detection performance and PI quantification accuracy. In total, four different validation metrics were used. F1 scores, which consider both sensitivity and precision of nuclei detection was used for Ki67+ and Ki67− nuclei detection performance (separately) with
(6)$$F1=\;\frac{2\times \mathrm{TP}}{\left(2\times \mathrm{TP}+\mathrm{FP}+\mathrm{FN}\right)}$$
where TP, FP and FN are the true positives, false positives and false negatives respectively. The images of binary predicted, and ground truth images are overlaid to examine whether each automatic nuclei detected corresponds to a manually labeled nuclei resulting in a true positive (TP) if the labeled seed is coincident with the predicted seed, an FP if the ground truth nuclei was missed and a FN if a nuclei was predicted where there wasn’t one in the ground truth image. These values are then used to generate an F1 score for each image in the dataset and to comparatively analyze across different experiments, the average F1 score is obtained per cohort.

To measure PI estimation accuracy, Proliferation Index (PI) difference is computed, the Pearson correlation coefficient to measure the consistency of PI quantification and the accuracy rate of PI classification into Low, Medium and High PI ranges. PI difference is used to investigate the overall error between predicted ($PI_{auto})$ and actual PI ($PI_{manual}$) value as shown in Equation (7). If a value for PI is given (SMH WSIs), the automatic PI and ground truth PI are compared. If the ground truths of a given dataset is marker-based, the PI is calculated using the markers, per stain, Equation (2) to find $PI_{manual}$ which is compared to the automated value $PI_{auto}$. The PI difference is computed as follows:(7)$$\Delta \mathrm{PI}=\left|PI_{manual}-PI_{auto}\right|$$

Pearson’s correlation coefficient (R) was applied as a validation metric to analyze the model’s ability output PI values that are consistent and correlated with the ground truth data. This metric is used to measure the linear association between the predicted PI ($PI_{auto}$) and actual ground truth PI ($PI_{manual}$), across a given cohort or dataset for analysis and is computed by:(8)$$R=\frac{\sum \left(PI_{auto}-\bar{PI_{auto}}\right)\left(PI_{manual}-\bar{PI_{manual}}\right)}{\sqrt{\sum {\left(PI_{auto}-\bar{PI_{auto}}\right)}^{2}\sum {\left(PI_{manual}-\bar{PI_{manual}}\right)}^{2}}}$$
where $\bar{PI_{auto}}$ represents the mean value of the predicted PI and $\bar{PI_{manual}}$ corresponds to mean value of the actual PI values.

A final measure known as the PI Accuracy Rate is also computed that considers whether the method predicts accurately in the respective PI ranges of Low, Medium, and High. The ranges used in most experiments is Low (<10), Medium (10–30), High (>30) as these ranges are common in the literature [10]. For experiments that include Protein Atlas we use PI ranges of Low (<25), Medium (25–75), High (>75) as these are the ground truth labels that are supplied. To measure PI range accuracy, if the range of the predicted PI value falls into the correct range this is counted as a correct classification. The total number of correctly classified PI ranges are accumulated and divided by the total number of data samples per cohort.

In addition to F1 scores, the difference between the methods can be elucidated in the PI Accuracy Rate and R which informs as to how well the classifier is performing over different PI ranges, as a function of disease severity. F1 scores report raw nuclei detection performance, and if the model is operating on an image with low numbers of tumor nuclei and a single nucleus is missed, this can greatly skew the overall F1 score of the patch allowing for outliers. Therefore, additional metrics, such as the PI Accuracy rate and R are needed. All four validation metrics will be analyzed together throughout the paper.

### 3.2. Model Development

Hyperparameter tuning is a critical component of model development. The two major concepts investigated here are, the choice of the regression-based loss function and the optimal dataset partitions between ideal and nonideal patches. SMH patches are used for testing, F1-scores validation metrics and PI agreement measures are used to analyze design combinations. PI range accuracy uses the proliferation index ranges of Low (<10), Medium (10–30) and High (>30) values. Pearson Coefficient (R) values are used to examine correlation between automatically measured PI values versus the manual ground truth PI. Throughout the course of the model development, the following experiments will be conducted to tune the detection model for optimal results: (i) loss function analysis across four functions and the best loss is used for the (ii) data partition experiment, where variations of ideal to nonideal patches will be tested to demonstrate the adaptability of the model.

#### 3.2.1. Loss Function

In deep learning, loss functions can enhance the performance of a model significantly, hence the evaluation of selecting a function which can advance the system is vital. As mentioned previously, piNet detects Ki67+/− tumor cells using Gaussian proximity maps in cell centers and therefore, a regression-based loss function is needed to optimize the model. Here we measure the performance of the presented loss functions: (i) Log Cosh, (ii) Huber Loss, (iii) Mean Square Error and (iii) Root Mean Square Error.

The model was trained using the 70:10:20 (training, testing and validation) data split on stratified data of equal distributions of Low (<10), Medium (10–30) and High (>30) PI expressions. A total of 330 ideal ROIs from the SMH patches dataset were used, 240 for training, 30 for validation and 60 ROIs, unseen by the model, for testing analysis. Holding all other hyperparameters consistent, the performance over each loss function was tested on same 60 SMH patches. The training dataset, 240 ROIs, remained consistent and without augmentations as well. Example detection results are shown for all loss functions, alongside the corresponding ground truths and original images in Figure 8. As shown, the tumor nuclei have been accurately demarcated over most loss functions with some slight variability in the results depending on the loss function used.

To analyze the performance of piNET quantitatively, validation metrics are computed over the testing dataset. The box plots for the average F1 scores for each of the tumor cell classes (Ki67+/−) are shown in Figure 9 for each of the loss functions. As can be seen, over all loss functions the F1 scores of piNET are mostly high with the negative class (Ki67−) with better performance than the positive Ki67+ class (Ki67− class has higher means, lower variance, and fewer outliers). The poorer performance of the Ki67+ class can perhaps be attributed to larger cell and stain variability, or less training data for this class. Ki67+ nuclei detection performs the worst with the MSE loss (high variance of F1 scores and extreme outliers). To examine the performance more closely, we summarized the average F1 scores for each tumor cell (Ki67+/−), overall accuracy rate and R for the four regression loss functions in Table 3. As shown, all the proposed loss functions accurately detect nuclei centers and differentiate between the two stains; all achieved an average F1 score above 0.8, except for MSE. RMSE achieved the highest Accuracy Rate and R, with competitive F1 scores for both classes. For these reasons, the RMSE is the leading option for the Gaussian-based tumor nuclei detection model and is selected for piNET. The scatter plot with corresponding R for all loss functions are shown in Figure 10.

#### 3.2.2. Data Partitions

In Ki67 breast cancer images, there are various types of tissue (adipose, stromal), epithelial cells and different artifacts such as folds, which can create false positives in nuclei detection [19,20,21,50]. Furthermore, for a robust PI quantification tool, non−tumor cells should not be included in the final PI calculation. Therefore, in this experiment, we introduce the concept of an addition of weakly supervised images, with minimal ground truth annotations. Weakly supervised labels consist of patch−based labels of “ideal” and “non−ideal”. Recall non−ideal patches could have some tumor cells, but they also largely have folds, artifacts, other tissue types and non−tumor cells. Nonideal patches are used to investigate whether the model can use this data to improve nuclei detection accuracy and reduce false positives. The goal is to let the classifier see varying partitions of these non−ideal patches of data, in order to learn these features as part of the background class. This could potentially improve the nuclei detection performance with the addition of weakly supervised images. An example of ideal (comprised of largely tumor tissue and cells) and nonideal patches are shown in Figure 7.

The 660 patches from SMH are comprised of 330 ideal patches and 330 nonideal patches. Throughout the experiment, the 240 training, 30 validation and 60 testing images of ideal patches do not change. Ideal data are stratified in terms of images of low, medium, and high PI (as per the loss function experiment). The primary addition are the non−ideal patches, while ideal ROIs remains consistent throughout analysis. In other words, the experiment will involve the addition of weakly labeled images which are nonideal patches of data, including, artifacts, folds, mixture of tissues. This allows for minimal ground truth generation, while remaining consistent in teaching the models contextual and spatial information. Ideal and non-ideal data partitions are split into varying percentages, 100:0, 70:30 and 50:50, each partition will be investigated. The first datasplit is 100:0, which consists of 100% ideal patches and no non-ideal images. The 330 ideal patches are then split into 70:10:20 sets for training, validation, and testing. The second datasplit is 70:30, which contains 30% nonideal patches resulting in 330 and 142 non-ideal patches, respectively. The last partition is 50:50 and all 330 ideal patches are used, and all 330 non-ideal patches are used. The number of images per stratification and datasplit for each experiment are listed in Table 4.

For the data partition experiments, the proposed piNET framework with the RMSE loss is trained and validated on the 70:10:20 datasplit for each partition, and testing is completed on 60 held out ideal patches. Example nuclei detection results for three ideal: nonideal ROI data splits (100:0, 70:30 and 50:50) are shown in Figure 11.

Figure 11 visually illustrates the importance of different data splits for analyzing images with large amounts of variation. The figure represents the models’ response to nonideal patches in the training sets. Using the 100:0 datasplit, it is obvious the nuclei detection phase is working well in the artifact free images (a–d). However, in images with overstaining, tissue folds or non−tumor nuclei (e–g) the classifier is detecting many false positive Ki67+/− nuclei. As nonideal patches of data are introduced in the 70:30 data partition, the model detects tumor nuclei but is also correctly suppressing overstaining, folds, and other non-tumor calls as part of the background class (e–g). Similar results are seen with the 50:50 partition. This demonstrates the importance of including variability in the background class, as it improves overall model performance.

The validation metrics are summarized in Table 5, the boxplots of the F1 scores are shown in Figure 12 and correlation between manual and automatic PI are illustrated in Figure 13. The F1 Score of Ki67+/− of the 70:30 data partition preforms the worst out of the three splits, specifically in the Ki67+ channel (DAB). The 100:0 and 50:50 data partitions have comparable F1 scores for both tumor cells, 0.9 for Ki67− and 0.8 for Ki67+, but it is evident from the visual results that artifacts, nontumor cells/tissue and folds would create challenges in the 100:0 datasplit. Perhaps this is reflected in the PI Accuracy Rate, which shows that the 50:50 ratio classifies the patches correctly, obtaining an accuracy rate of 93% on classifying PI within the Low (<10), Medium (10–30) and High (>30) ranges. This may be attributed to improved classification of folds, artifacts, stromal tissue and other cell/tissue types as background (as shown in Figure 11) Therefore, to minimize false positives, and to obtain desirable detection performance, the 50:50 data partition of ideal and nonideal patches is used.

### 3.3. Nuclei Detection Accuracy and Generalization to other Datasets

Using a 50:50 data partition of ideal-nonideal training patches and RMSE loss function for piNET, the overall nuclei detection accuracy and performance on unseen datasets of varying scanner vendors, image types (ROI, TMAs, WSIs), laboratory protocols or biomarker expression levels will be examined. In addition to reporting metrics on the held out SMH patches, DeepSlides (human breast tissue), and the OVC TMAs (canine mammary) will be investigated using F1 scores (since these datasets have individually annotated nuclei). See Figure 14 for the box plots for the F1 scores for each of the datasets, and for each tumor nuclei type separately. The average metrics per dataset are shown in Figure 15a and the correlation between the computed and ground truth PI are shown in Figure 15b. The average metrics are summarized in Table 6.

As shown by the F1 scores, the highest performance of the model is computed on the held out SMH patches. Similar performance is noted for DeepSlides although the Ki67+ detection accuracy was higher in DeepSlides than in the SMH dataset, this is interesting since data from this cohort was not seen in training. The data was also down sampled (to 20×) and despite this, the model generalized well to this dataset. The OVC data had the lowest F1 scores, given the TMAs consisted of canine mammary microstructure, there can be some performance drop expected-due to differences in cells, but overall, the model is still performing well. The correlation between manual PI and automatic PI of the various datasets were calculated using R and the PI Accuracy Rate were also computed. All three datasets achieved a R value above 96%, demonstrating that the proposed piNET model can generalize to various unseen datasets. Interestingly, the OVC TMAs achieved the lowest F1 score and accuracy rate, but highest R, despite being canine mammary samples. There were some TMAs in the OVC dataset with lower PI values which the model had missed, this brought the overall F1 score down. However, when looking at the correlation between manual and automated PI calculation, there is clear indication that the model can predict PI accurately (R = 0.982).

### 3.4. Proliferation Index Accuracy on Diverse Data

Previously, piNet was tested on individually annotated nuclei to analyze the overall model’s ability to detect nuclei across three datasets, while only trained on a single dataset. In this section, piNET will be applied to all datasets mentioned in Table 1 and the PI accuracy will be measured with Proliferation Index Difference, Pearson correlation coefficients and PI Accuracy Rate. Of the five datasets, four provide proliferation index values, while Protein Atlas uses proliferation index range classifications (Low < 25, Medium 25–75, High > 75). Hence, four datasets will be analyzed using PIs directly (PI difference and correlation coefficients) and all five datasets will be analyzed using the PI range classifications of Protein Atlas. There are three types of images included in this analysis: TMAs, ROIs and WSIs. TMAs and ROIs are generally easier to process than WSIs since they are cleaner data, meaning artifacts folds are minimal or nonexistent and the majority of the image contains tumor cells. WSIs typically are complex to quantify and difficult to analyze due to the variability within a single slide (various cell types, tissues, and artifacts) and is a realistic illustration of clinical data pathologists work with.

To visualize results within the dataset, Figure 16 illustrates some nuclei detection results and the corresponding computed PI across 4 datasets. SMH patches are perceptively different to DeepSlides for example, the slides are stained with a purple-ish hematoxylin, darker DAB stains and contain larger tumor cells. The Protein Atlas TMAs vary in staining color, scanning vendor, and the overall biomarker expression levels. These images largely consist of tumor regions, with minimal excessive nontumorous cells/tissue or artifacts. OVC TMAs, as previously mentioned, is canine mammary data. However, regardless of the image or tissue type, the model is able to accurately analyze the tumor cells effectively.

The average PI accuracy metrics are summarized in Table 7 over all five datasets using the PI difference, accuracy rate using Low < 25, Medium 25–75 and High > 75 PI ranges, along with the Pearson’s correlation coefficient between manual and automatic PI. The difference in PI over all datasets is relatively low. Both the patched datasets (SMH and DeepSlides) as well as the OVC TMAs obtain a PI difference below 5%, indicating that piNET is accurately quantifying PI. SMH WSIs achieves a PI difference of approximately 11%. Although this is higher than the three comparative datasets, WSIs are larger images to analyze and carry greater amounts of variation. A similar trend can be seen for the Pearson related coefficient values, where the three datasets consistently obtain a higher value and SMH WSIs achieves the lowest correlation. Analyzing WSIs is a challenging problem, due to the sheer size of the data and there are varying methods for analysis [19,20,24]. Whole slide images can consist of 100,000 × 100,000 pixels [51]. Despite the large amount of data that is processed, our model is able to achieve a 11% PI difference between the pathologist’s representative regions approach and the automated piNET method.

Across the five datasets, three image types, four laboratory sources, the piNET model is able to achieve an PI accuracy rate of 85.2%, while being trained on data from a single source. The accuracy rate of the two ROI datasets (SMH and DeepSlides), combined, on average is 88% with PI difference of 4.23% and a R of 97%. The two TMA datasets obtained a high PI accuracy rate: 97% on the OVC dataset, which is non-human breast cancer data, and 80% on the Protein Atlas dataset, both unseen by the model. The proliferation index difference of the WSI dataset is approximately 11%, with an accuracy rate of 76% and a PI Pearson related coefficient of 68%. The WSI dataset scores the lowest of the 5 datasets because of how the proliferation index ground truth values were obtained. The computed difference in PI versus ground truth values across the four datasets can be seen in Figure 17.

### 3.5. Comparison to other Works

In addition to PI accuracy over all datasets, we also compare the performance of the proposed piNET system to other traditional and deep learning models. The piNET framework was built based on the U-NET architecture and ResUNet [47], which is what KiNet [22] was built upon, DenseUNet [48] and FCN8 [49] are other deep learning architectures investigated. All models are a variation of the original U-NET convolutional neural network. Each model used the same learning rate (learning rate = 0.001), Adam optimizer, RMSE loss function over 100 epochs. The SMH training dataset, which was previously used in the other experiments, is used for training and testing. The deep learning methods are also compared to our previous work on the IHC Color Histogram (IHCCH) [18]. It is an unsupervised method that uses a novel color separation approach to analyze hematoxylin (Ki67−) and DAB (Ki67+) stains separately, and a gradient-based nuclei detection method to detect cells in each channel and quantify PI. Each one of the five datasets, for each method, were tested for generalizability using the PI accuracy measurements.

To examine the PI estimation accuracy for each approach, the manual and automatic PI were plotted in Figure 18 for all datasets combined. Of the four comparative methods, the piNET seems to be performing the best, with an average PI difference of 5.6% across four datasets (with an overall lower variance and outliers). This demonstrates that piNET can generalize to new datasets better than competing methods. Table 8 represents the average PI difference categorized per dataset, where you are able to see the proposed method achieves the lowest PI differences in comparison to the other models analyzed. The unsupervised method, IHCCH, obtained the lowest average PI difference on SMH WSI, but the highest on DeepSlides and the SMH patches, hence this method did not excel in the generalizability test.

The correlation between manual and automatically generated PI values per model and the Pearson correlation coefficient were calculated, per dataset, are shown in Table 9. The proposed method, piNET, is able to obtain the most consistent R value over all datasets, including the WSIs. However, the SMH WSIs values are lower than other datasets due to the variability within the WSI. Breast cancer disease expresses heterogenous characteristic traits, where there is a high degree of variability within the tumor’s pathology. This can be seen in Figure 3, where a WSI is annotated to illustrate the various levels of biomarker expression within a single image. FNC8, ResUNet and IHCCH method all perform well, but when analyzing the WSIs, piNET outperforms the others.

Finally, accuracy rate, Table 10, of each image classification of Low (PI < 25), Medium (PI = 25–75) and High (PI > 75) were analyzed. The accuracy rate, as mentioned, is a proportion of the models’ ability to classify the PI ranges precisely across the data sample. This was conducted across all five datasets, and overall, piNET outperforms the rest. The proposed method’s ability to achieve the highest value consistently throughout each dataset. Hence, piNET is able to generalize to unseen data and is the optimal configuration for automated tumor nuclei detection for Ki67 stained breast tumor data.

## 4. Discussion

In digital pathology, there are a variety of challenges that algorithms must overcome when analyzing multi-institutional data, a large challenge being variability. The datasets analyzed in this work vary in, stain vendors, antibody biomarkers expression levels, laboratory staining protocols, different scanner vendors, scanner magnifications and image types (TMA, WSI, ROI). When using neural networks as a tool for biomedical images, the issue of overfitting to a single dataset is a concern. The model may learn features from a single institution that do not generalize to other datasets from different laboratories. The proposed pipeline was implemented with robustness and generalizability in mind, using five datasets for validating these critical challenges. The multi-institutional dataset consists of four sources, three types of images, three scanner models and two types of mammary tissue (human and canine). An overall accuracy rate of 85.2% across five datasets was achieved, an average of 87.9% across ROIs, 85.5% for TMAs and the single WSI dataset obtained 76.4%, using the Low < 25, Medium 25–75 and High > 75, PI classification. To examine the WSI performance, outlier analysis was performed using the z-score. A single outlier was identified, and if removed from the SMH WSIs analysis, the PI difference changes from 10.997% to 10.460% and R increased from 0.684 to 0.75. Figure 19 illustrates the scatterplot of the automatic versus manual PI with the exclusion of a single outlier.

Analysis of WSIs face different challenges, in comparison to ROIs and TMAs since biopsy slides tend to have variations of tissue types, artifacts, folds and different cell types (tumor and nontumor). When training a neural network model, if the patches that are being trained are “ideal”, i.e., only tumor tissue and cells (exclusion of artifacts), the model may fail to generalize in WSI data. To overcome this, a novel concept of data partitions or ideal/nonideal labels (weak supervision) were introduced. As was shown, the data partition of 50:50 outputs F1 scores within the same range as the ideal data partition of 100:0. However, the 50:50 data partition has reduced false positives in the presence of folds, artifacts and nontumor cells, indicating the system learnt these features as part of the background class. Figure 11 illustrates the model’s ability to generalize to artifacts, stromal tissue, and folds within WSIs, using the addition of weakly annotated images, experimenting with three different data partitions. These ROIs were cropped from SMH WSIs, and the first four images (a–d) illustrate a mixture of ideal and nonideal patches of data and the last three (e–g) represent nonideal ROIs which are frequently found in the digital pathology images. Visually it is easy to see that the model trained on solely ideal patches (100:0), preforms poorly with many false positives when folds and artifacts are present. The model trained on a mixture of ideal and nonideal can adapt to realistic, clinical data (WSIs), which consists of folds, artifacts and overstained regions.

Due to variations in biomarker expression, tumor heterogeneity and the WSIs’ large size, analyzing the entire WSI is not time efficient for pathologists. Automated methods can be used to alleviate some of these burdens and for WSI validation, the automated PI must be compared to the pathologist (expert) PI. The ground truths for the WSI dataset were obtained by a breast pathologist who used representative regions to obtain the PI score. The pathologist observes the overall biomarker expression throughout the tumor and focuses analysis on three regions, which consists of a minimum of 500 cells, that are representative of the overall tumor. These representative regions are used to compute the final whole slide PI. In this work, we propose an innovative method of automatically calculating hotspot regions, in order to provide a PI score. While we do not only use three regions, this method finds patches that have significant activity and use the cumulative tumor cell counts to measure whole slide PI. Using this approach, there was a PI difference of 10.997%, on the WSI dataset and represents a low error rate and an accurate quantification without the known regions which the pathologist had analyzed. In the future, upon integration to whole slide imaging viewers, we will consider scoring only regions that are selected by pathologists.

When analyzing the comparative methods, piNET achieves the lowest average PI difference across all datasets, with optimal results for DeepSlides and OVC TMAs. However, it was slightly outperformed by Res-UNet on SMH patches and by the unsupervised IHCCH pipeline on the WSI data (0.5% difference). The benefit of the current approach is that it is easy to change architectures (Res-NET, Dense-NET, etc.,) and see which is optimal for different datasets, which is a subject of future works. Considering the Pearson related coefficient and classification accuracy rate on the WSI, the piNET surpasses the IHCCH method, where piNET achieves R = 0.773 and an accuracy rate of 76.4% while IHCCH obtains R = 0.578 and a rate of accuracy of 72.7%. Therefore, although the PI differences were slightly higher on the IHCCH method, piNET is demonstrating robustness over all PI ranges through correlation (R) and PI accuracy. Furthermore, piNET has higher accuracy (generalizes) across all five datasets and the comparative models do not generalize as well to multi-center data.

The preprocessing steps used for TMAs and WSIs eliminates the need for users to annotate regions of interest. When analyzing TMAs, specifying regions is not necessary, since TMAs are relatively clean datasets, and the entire region can be used for quantification. As shown in Figure 2, similarly both Protein Atlas and the OVC TMA datasets are visually very different looking, the stains and overall nuclei shape vary from the SMH dataset, which the dataset which the model was trained upon. Protein Atlas consists of ground truth values which are PI classification ranges of Low (<25), Medium (25–75) and High (>75) and the OVC TMAs ground truths are PI values, which can be classified into ranges. The proposed method will quantify PI of the inputted TMAs, which will then be classified based on the predicted PI value. A total of 60 Human Breast tissue TMAs from the Protein Atlas dataset were used for validation, each of these TMAs varied in image size, staining intensity, biomarker expression and antibody types (HPA000451, HPA001164, CAB000058, CAB068198). From the 60 TMAs the proposed method correctly classified 48 images, achieving an accuracy rate of 80.0%. The OVC dataset consists of 30 canine mammary tumor TMAs, of which 29 were correctly classified, obtaining an accuracy rate of 96.7%. Overall, the TMA dataset, combined with Protein Atlas and OVC, composed of 90 TMAs, a total of 77 were precisely classified, obtaining 85.6% accuracy rate. Figure 16 illustrates six images of the ground truth classifications or PI values with the original TMAs from Protein Atlas (g–i) and OVC (j–l), underneath you can visually see the overlaid predictions on each image with the corresponding classification or PI value. The model’s ability to ignore stromal tissue, epithelial cells and other non-tumor cells, while only examining the tumor cells of Ki67+/− is illustrated.

To demonstrate proof of concept and clinical utility, we have also started looking at automated WSI PI scores and comparing them to PHH3 counts. PHH3 is a mitotic activity marker, and there is interest in finding relationships between proliferation and mitotic activity. Ki67 stains nuclei of all active phases within the cell cycle, i.e., G1, S and G2 phases. However, PHH3, phosphohistone H3, actively stains to mitotically active cells, which is also proliferation explicit, specifically in the late G2 and metaphase stages [52]. In [53], a strong linear correlation was found between Ki67 and PHH3 in Ductal Carcinoma In situ (DCIS). In this work, based on invasive breast cancer disease, manual PHH3 counts were correlated with the automated Ki67 PI scores for the SMH WSI data. The PHH3 counts were acquired by an experienced breast pathologist, who evaluated 10 high power fields on adjacent tumor slides to the Ki67 stained tissue sections, with a field diameter of 0.65 mm, per corresponding PHH3 WSI. Comparing piNET’s automated PI score to the PHH3 index, a correlation of R = 0.57 was obtained as shown in Figure 20. Comparing the three supervised methods and the unsupervised IHC Color Histogram-based method, the piNET is able to achieve the highest correlation coefficient. Comparing the manual PI and PHH3 scores, a correlation value of R = 0.7425 was achieved. This may be due to the fact that the same regions were scored in both the Ki67 and the PHH3 slides by the pathologist. In comparison, the automated method (piNET) used all hotspot regions that had significant activity. In the future we will investigate further the impact of the hotspots selected. In [52], the study concluded that digital image analysis tools to analyze PHH3 and Ki67 stains, in combination with Mitotic Activity Index (MIA), has a strong prognostic value. The use of digital image analysis for Ki67 and PHH3 or MIA, had stronger prognostic insight in contrast to Ki67 alone. Overall, the use of DIA in digital pathology has opportunities to improve patient quality of care, predict prognostic outcomes and survival rates.

Digital image analysis is a great tool to aid pathologists in their everyday workflow. It can relieve them of stressful tasks, overall workload, and the speed of analysis can improve patient quality of care. As artificial intelligence becomes more prominent in the biomedical space, in future works, we plan to minimize the piNET’s error rate by considering novel stitching methods, investigating the impact of Otsu’s threshold on prediction accuracy and exploring data augmentation. The use of Generative Adversarial Networks (GANs) to add diversity to the training dataset to improve prediction, as well as post processing methods to minimize error rates will also be analyzed.

## 5. Conclusions

The proposed framework is a fully automatic Ki67 Proliferation Index Calculator, piNET, which calculates the proliferation index (PI) over five datasets from four sources, three image types (TMAs, patches and WSIs) and three scanner types, with varying color and stain characteristics. The pipeline was trained on a single dataset, and when tested across multi-center data. It achieved an average PI difference of 5.603%, correlation coefficient R = 0.927, and an overall accuracy rate of 85.2% across five datasets; an average of 87.9% across ROIs, 85.5% for TMAs and the single WSI dataset obtained 76.4%, using the Low < 25, Medium 25–75 and High > 75, PI classification. The pipeline’s PI difference, Pearson related coefficient and accuracy rate in combination with five datasets outperformed state-of-the-art methods. The success of the system can be attributed to several novelties and innovations in the system. Firstly, this tool is built based on deep learning, which can adapt to wide variability of medical images–and it was posed as a detection problem, instead of a segmentation one to mimic pathologists’ workflow which improves accuracy and efficiency. Secondly, the system is trained purely on tumor cells, which avoids a usual necessary pre-requisite tumor segmentation step for Ki67 quantification, which helps to reduce inflated PI estimates and false positives from non-tumor cells. Thirdly, we introduce the concept of learning background regions through weak supervision, by providing the system with ideal and non-ideal patches which contain folds, artifacts and non-tumor cells and tissue. Lastly, we provide a novel hotspot analysis that allows automated methods to only score patches from the WSI that contain “significant” activity.

## Figures and Tables

**Figure 1 cancers-13-00011-f001:**
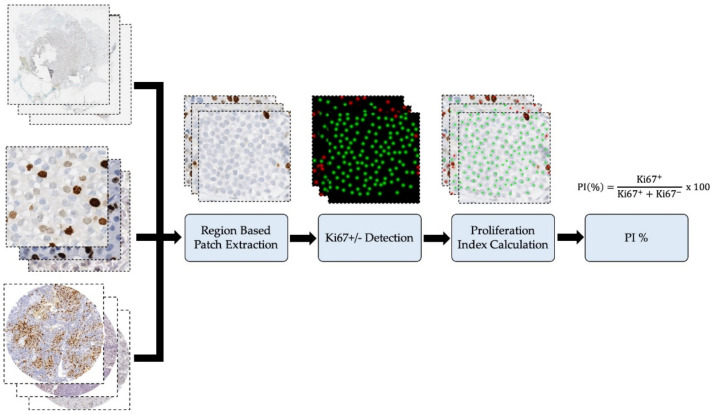
Proposed piNET framework, a fully automated Ki67 proliferation index calculator.

**Figure 2 cancers-13-00011-f002:**
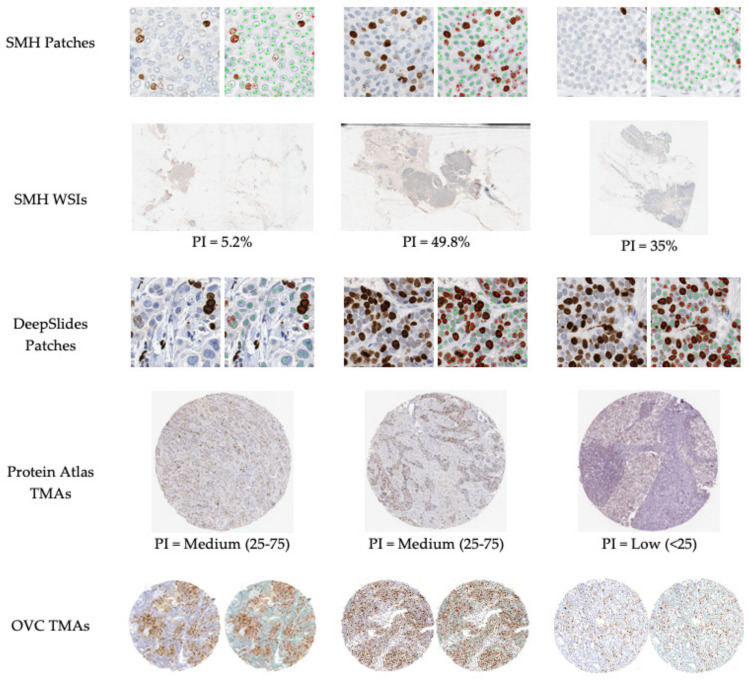
Example images from the datasets used in this work with corresponding ground truths. There is clear variability in the types of images, staining concentrations and color variability in the datasets.

**Figure 3 cancers-13-00011-f003:**
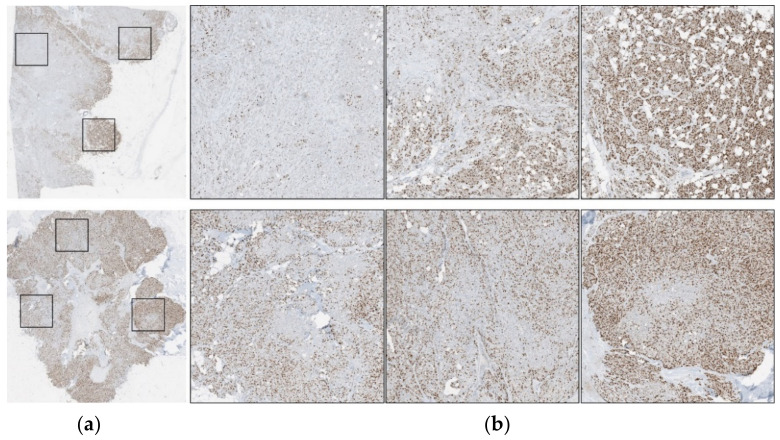
Variability of a WSI can be displayed in (**a**) W SI images, with corresponding (**b**) candidate regions for analysis.

**Figure 4 cancers-13-00011-f004:**
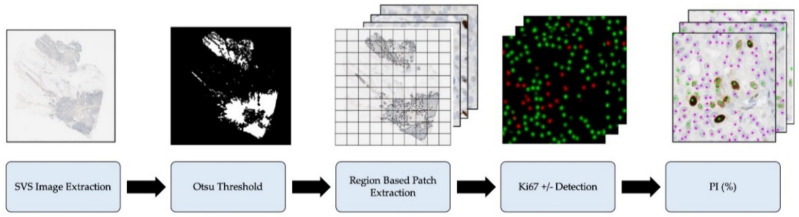
WSI Proliferation preprocessing steps, prior to Ki67+/− detection and PI quantification.

**Figure 5 cancers-13-00011-f005:**
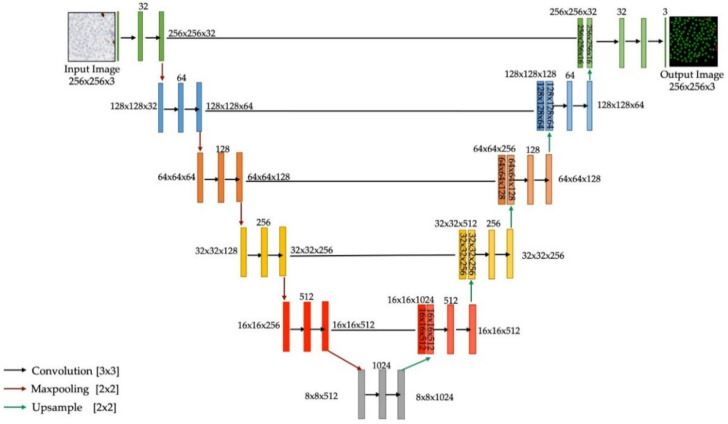
Ki67+/− detection is preformed through a U-NET model.

**Figure 6 cancers-13-00011-f006:**
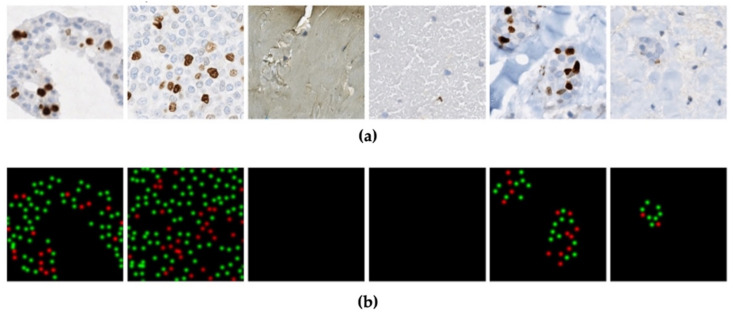
Array of ROIs from the SMH dataset. (**a**) Sample patches (ROI), (**b**) corresponding tumor cell proximity maps (red is Ki67+, green is Ki67− and black is the background).

**Figure 7 cancers-13-00011-f007:**
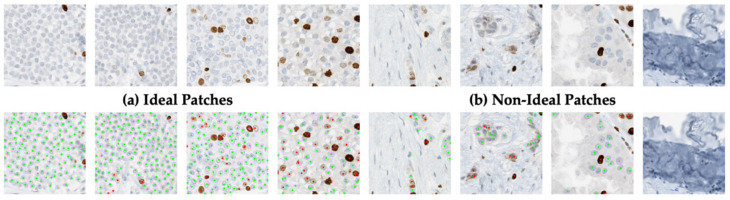
Array of SMH patches for (**a**) ideal tumor patches with majority tumor cells and (**b**) nonideal patches, filled with folds, stromal tissue, epithelial cells, inflammatory cells and combination of tumor/nontumor cells/tissue.

**Figure 8 cancers-13-00011-f008:**
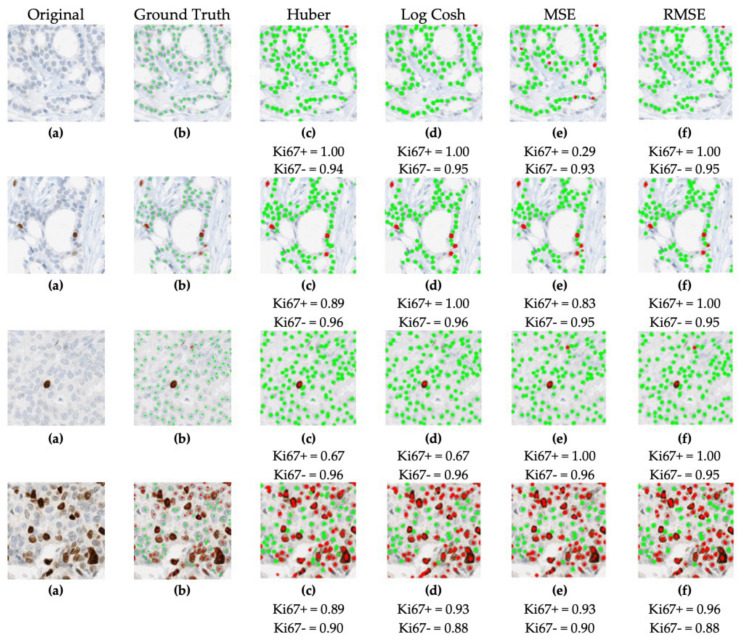
Visual representation of four patches of the (**a**) original image and corresponding (**b**) ground truth annotations alongside each loss function trained model’s output, (**c**) Huber, (**d**) Log−Cosh, (**e**) MSE, (**f**) RMSE.

**Figure 9 cancers-13-00011-f009:**
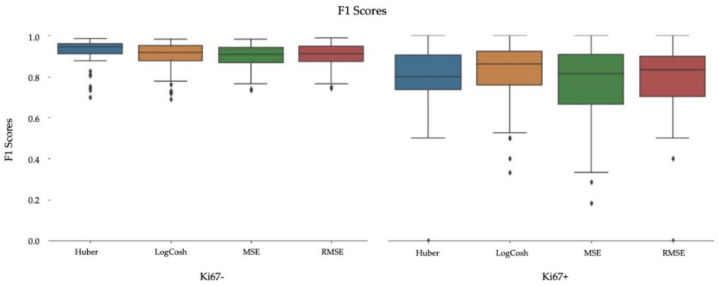
F1 scores on the SMH patches dataset, per stain Ki67− (Hematoxylin) and Ki67+ (DAB).

**Figure 10 cancers-13-00011-f010:**
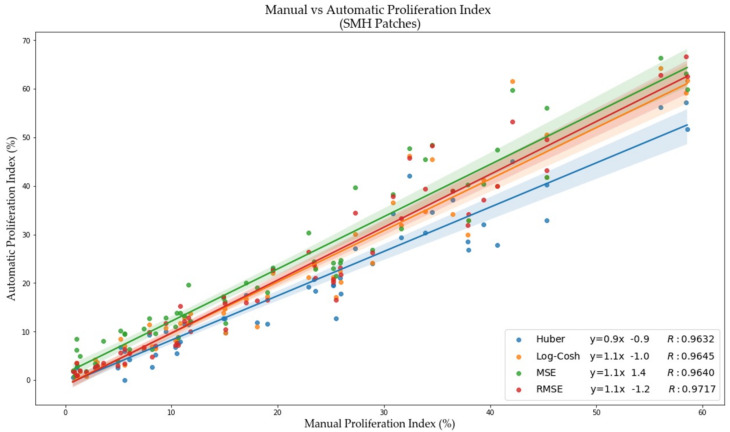
PI scatter plot of manual PI vs. automated PI for each regression-based loss function, with the corresponding Pearson Coefficient (R), and equation of the line.

**Figure 11 cancers-13-00011-f011:**
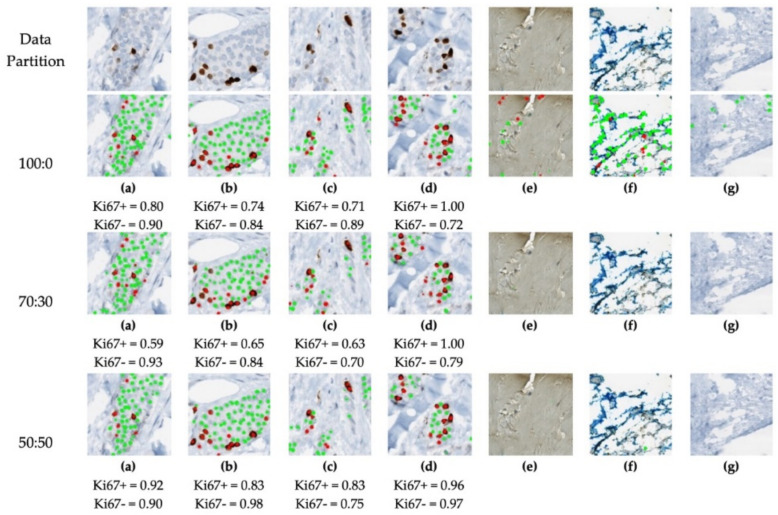
Nuclei detection performance of piNET trained using different amounts of nonideal patches of data. (**a**–**d**) images contain tumor nuclei and (**e**–**g**) contain background regions with no tumor nuclei. Green annotations correspond to hematoxylin labels and red to Ki67+ labels.

**Figure 12 cancers-13-00011-f012:**
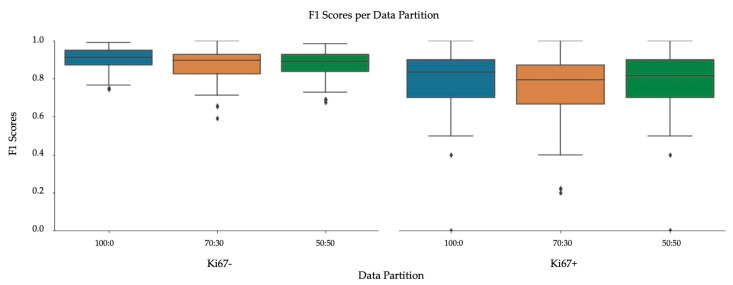
F1 Scores on SMH patches, per stain (Ki67− and Ki67+), comparing data partitions.

**Figure 13 cancers-13-00011-f013:**
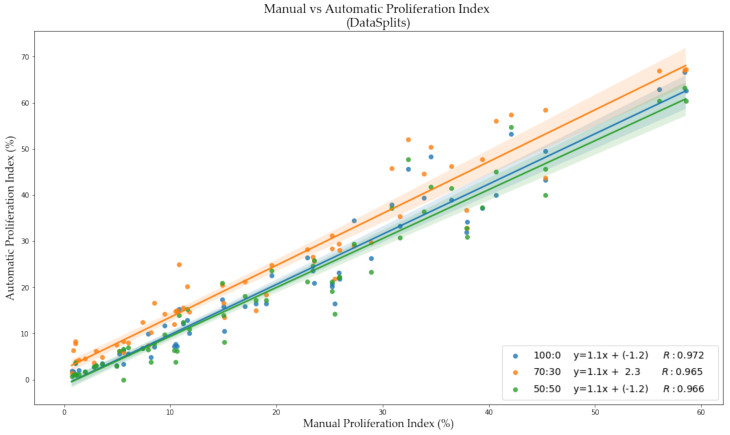
PI scatter plot of manual PI vs. automated PI for each data split experiment, with the corresponding Pearson Coefficient (R), and equation of the line.

**Figure 14 cancers-13-00011-f014:**
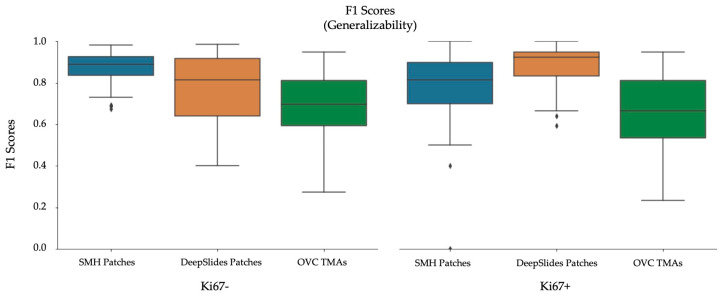
F1 scores on three datasets, SMH patches, DeepSlides, OVC per stain (Ki67− and Ki67+).

**Figure 15 cancers-13-00011-f015:**
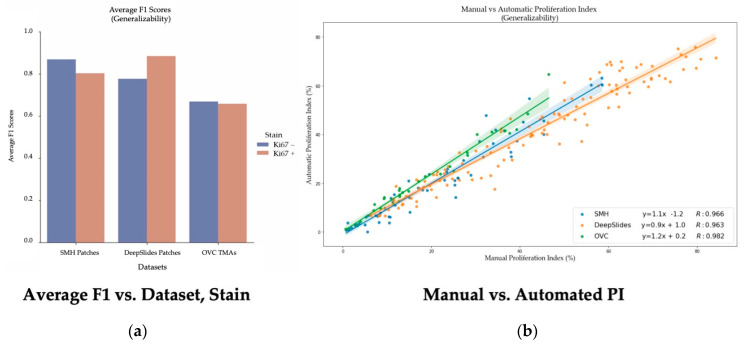
(**a**) Average F1 scores and (**b**) Manual vs. Automatic PI on three datasets: SMH Patches, DeepSlides and OVC TMAs.

**Figure 16 cancers-13-00011-f016:**
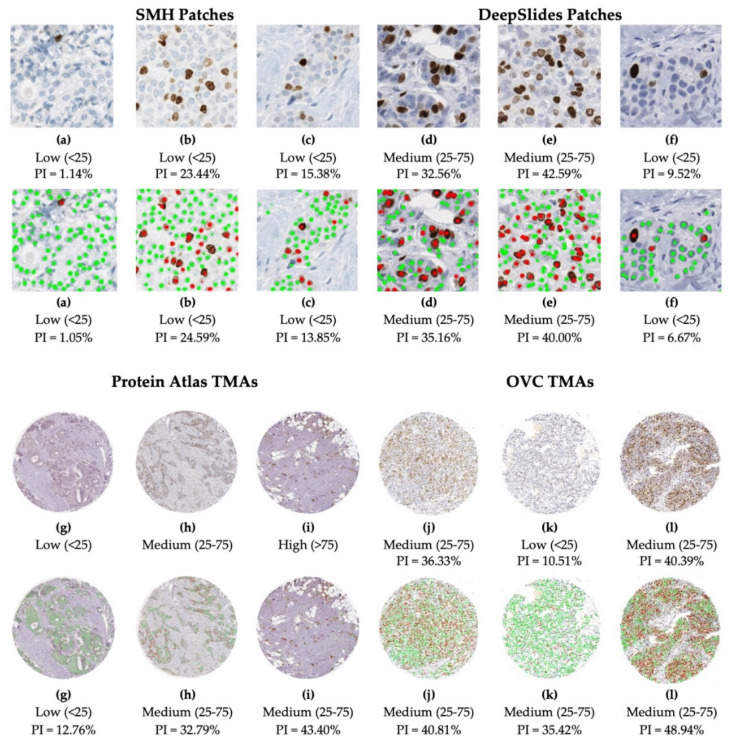
piNet, trained on SMH patches (**a**–**c**); model’s ability to generalize to three unseen datasets, DeepSlides ROIs (**d**–**f**), Protein Atlas TMAs (**g**–**i**) and OVC TMAs (**j**–**l**). A visual representation of the variation of data within datasets. Original images (top rows) with corresponding ground truth PI values and ranges. piNet predicted annotated images (bottom rows) along with the automatically calculated PI and classification range.

**Figure 17 cancers-13-00011-f017:**
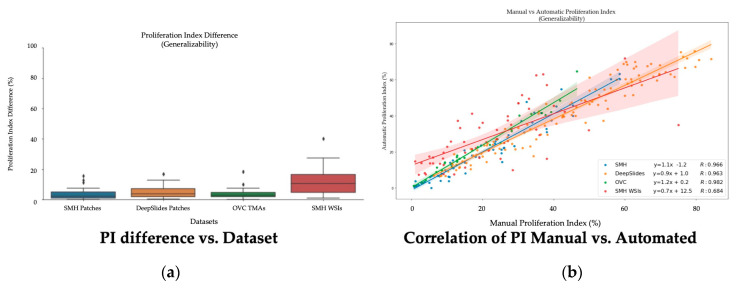
(**a**) PI difference and (**b**) manual vs. automated PI over all four datasets.

**Figure 18 cancers-13-00011-f018:**
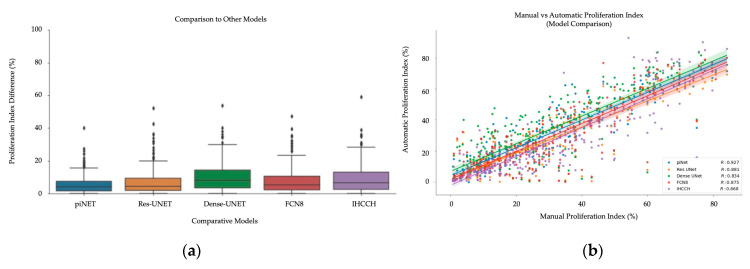
(**a**) PI difference of piNET over four datasets compared to three deep learning methods (ResUNet, DenseUNet, FCN8) and an unsupervised method (IHCCH). (**b**) Correlation between automated and manual PI overall datasets and for each method.

**Figure 19 cancers-13-00011-f019:**
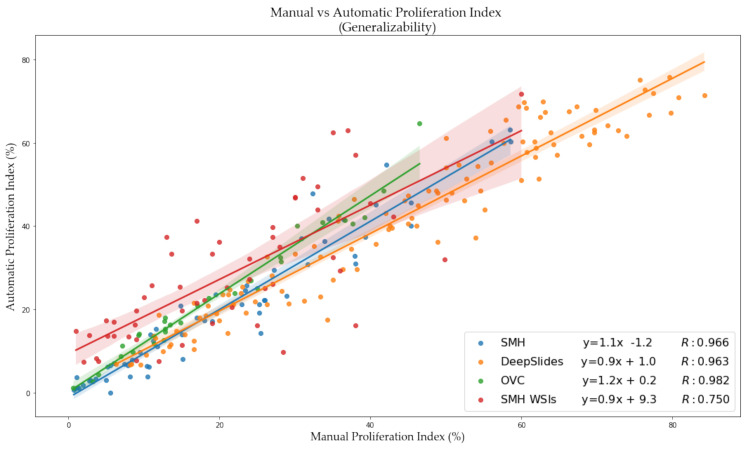
Correlation of automated and manual PI estimates over four, multi-centre datasets with single outlier removed.

**Figure 20 cancers-13-00011-f020:**
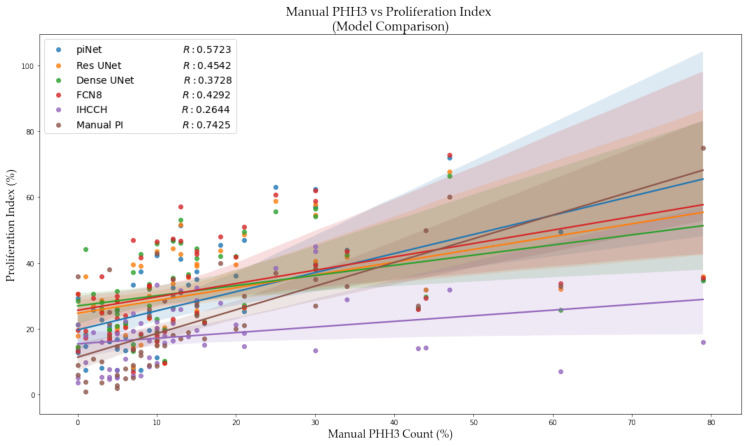
Scatter plot of PHH3 mitotic count versus Ki67 proliferation index (%) for the pathologist ground truth PI and the automated PI results from proposed pipeline and the comparative methods.

**Table 1 cancers-13-00011-t001:** Ki67 breast cancer digital pathology datasets used to evaluate piNET.

Dataset	Source	Image Type	Ground Truth	Quantity	Scanner	Mag	Tissue
SMH Patches	St. Michael’s Hospital	Patches	Individual Nuclei	Ideal: 330 Non Ideal: 330	Aperio AT Turbo	×20	Human Breast
SMH WSIs	St. Michael’s Hospital	WSI	Proliferation Index	55	Aperio AT Turbo	×20	Human Breast
Deep Slides	Senaras et al. (Open Source)	Patches	Individual Nuclei	452	Aperio ScanScope	×40	Human Breast
Protein Atlas	Protein Atlas (Open Source)	TMA	Proliferation Index Ranges	56	Aperio AT Turbo & Aperio T2	×20	Human Breast
OVC	Ontario Veterinary College	TMA	Individual Nuclei	30	Leica SCN400	×20	Canine Mammary

**Table 2 cancers-13-00011-t002:** Regression loss function formulas.

Loss Function	Formula
Huber $\left(y,\;\hat{y}\right)$ =	$\left\{\begin{array}{ccc} \frac{1}{2}{\left(y_{i}-\;\hat{y_{i}}\right)}^{2}, & & when\;|y_{i}-\;\hat{y_{i}}|\leq \partial \\ \partial |y_{i}-\;\hat{y_{i}}|-\frac{\partial^{2}}{2}, & & otherwise \end{array}\right.$
LogCosh $\left(y,\;\hat{y}\right)$ =	$\overset{n}{\underset{i=1}{\sum }}log(cosh(\hat{y_{i}}\;-\;y_{i}))$
MSE $\left(y,\;\hat{y}\right)$ =	$\frac{1}{n}\overset{n}{\underset{i=1}{\sum }}{\left(y_{i}\;-\;\hat{y_{i}}\;\right)}^{2}$
RMSE $\left(y,\;\hat{y}\right)$ =	$\sqrt{\overset{n}{\underset{i=1}{\sum }}\frac{{\left(y_{i}\;-\;\hat{y_{i}}\right)}^{2}}{n}}$

**Table 3 cancers-13-00011-t003:** Average F1 scores, accuracy rates and R values of SMH patches per loss function.

Loss Functions	Avg F1 Scores		Accuracy Rate (Low < 10, Med 10–30, High > 30)	R (*p*-Value) (Manual vs. Automatic PI)
Huber Loss	Ki67− Ki67+	0.922 0.808	0.831	0.9632 (*p* < 0.05)
Log Cosh	Ki67− Ki67+	0.896 0.814	0.867	0.9645 (*p* < 0.05)
MSE	Ki67− Ki67+	0.899 0.763	0.881	0.9649 (*p* < 0.05)
RMSE	Ki67− Ki67+	0.901 0.803	0.917	0.9717 (*p* < 0.05)

**Table 4 cancers-13-00011-t004:** Number of images used for each data split partition on SMH ROIs.

Data Partition %Ideal: %Nonideal	Total # of Patches	Training, Validation, Testing Split (70:10:20)	Training Ideal: Nonideal	Validation Ideal: Nonideal	Testing Ideal: Nonideal
100:0	330	240:30:60	240:0	30:0	60:0
70:30	472	343:43:86	240:103	30:13	60:26
50:50	660	480:60:120	240:240	30:30	60:60

**Table 5 cancers-13-00011-t005:** Average F1 scores and accuracy rate across three data partitions.

Data Partition Ideal: Nonideal	Avg F1 Scores		Accuracy Rate (Low < 10, Med 10–30, High > 30)	R (*p*-Value) (Manual vs. Automatic PI)
100:0	Ki67− Ki67+	0.901 0.803	0.917	0.972 (*p* < 0.05)
70:30	Ki67− Ki67+	0.870 0.740	0.917	0.965 (*p* < 0.05)
50:50	Ki67− Ki67+	0.868 0.804	0.933	0.966 (*p* < 0.05)

**Table 6 cancers-13-00011-t006:** Average F1 scores and accuracy rates across three datasets of individually annotated nuclei.

Dataset	Avg F1 Scores		Accuracy Rate (Low < 10, Med 10–30, High > 30)	R (*p*-Value) (Manual vs. Automatic PI)
SMH Patches	Ki67− Ki67+	0.868 0.804	0.933	0.966 (*p* < 0.05)
DeepSlides Patches	Ki67− Ki67+	0.777 0.885	0.912	0.963 (*p* < 0.05)
OVC TMA	Ki67− Ki67+	0.669 0.659	0.833	0.982 (*p* < 0.05)

**Table 7 cancers-13-00011-t007:** PI accuracy of piNET on diverse data.

Dataset	F1 Scores		Avg PI Difference	R (*p*-Value) (Manual vs. Automatic PI)	PI Accuracy Rate (Low < 25, Med 25–75, High > 75)
SMH Patches	Ki67− Ki67+	0.868 0.840	3.237	0.966 (*p* < 0.05)	0.883
Deep Slides Patches	Ki67− Ki67+	0.777 0.885	4.749	0.963 (*p* < 0.05)	0.876
OVC TMAs	Ki67− Ki67+	0.669 0.659	3.662	0.982 (*p* < 0.05)	0.967
SMH WSIs	--		10.997	0.684 (*p* < 0.05)	0.764
Protein Atlas TMAs	--		--	--	0.800
Entire Dataset	Ki67− Ki67+	0.788 0.820	5.603	0.927 (*p* < 0.05)	0.852

**Table 8 cancers-13-00011-t008:** Comparison of average PI difference across four datasets.

Dataset	piNET	ResUNet	DenseUNet	FCN8	IHCCH Unsupervised
SMH Patches	3.237	2.776	3.211	3.160	6.303
DeepSlides	4.749	5.155	10.410	5.270	10.868
OVC TMAs	3.662	5.692	10.789	10.165	3.285
SMH WSIs	10.997	16.145	16.123	15.341	10.496
Entire Dataset	5.603	7.007	9.998	7.495	8.845

**Table 9 cancers-13-00011-t009:** Pearson related coefficient (R) across four datasets.

Dataset	piNET	ResUNet	DenseUNet	FCN8	IHCCH Unsupervised
SMH Patches	0.966	0.973	0.960	0.964	0.872
DeepSlides	0.963	0.962	0.918	0.956	0.895
OVC TMAs	0.982	0.971	0.890	0.963	0.932
SMH WSIs	0.684	0.580	0.519	0.559	0.405
Entire Dataset	0.927	0.907	0.888	0.896	0.868

**Table 10 cancers-13-00011-t010:** PI accuracy rate across all five datasets.

Dataset	piNET	ResUNet	DenseUNet	FCN8	IHCCH Unsupervised
SMH Patches	88.3%	90.0%	91.7%	93.3%	80.0%
DeepSlides	87.6%	85.8%	70.8%	86.7%	71.7%
OVC TMAs	96.7%	90.0%	83.3%	90.0%	93.3%
SMH WSIs	76.4%	69.1%	67.3%	74.5%	72.7%
PT Atlas	80.0%	61.7%	53.3%	43.3%	66.7%
Entire Dataset	85.2%	79.6%	72.0%	78.0%	74.5%

## Data Availability

The publicly available dataset that was analyzed in this study can be found here: https://www.proteinatlas.org/ENSG00000148773-MKI67/pathology/breast+cancer# and https://zenodo.org/record/1184621.

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
