# Peer review of "piNET–An Automated Proliferation Index Calculator Framework for Ki67 Breast Cancer Images"

_cancers, 2020, doi:10.3390/cancers13010011_

Round 1

Reviewer 1 Report

Review  cancers-1026564: piNET: An Automated Proliferation Index Calculator Framework for Ki67 Breast Cancer Images

I would like to congratulate the authors for their excellent work and implementation of a complex project.

The authors took the construction of a novel automated PI assessment system for KI67 in breast cancer using deep learning technology. The subject, material and methodology are valuable and very interesting.

The manuscript requires minor changes:

  • Introduction - all paragraphs are too long – need shortening
  • Line - 344 and 501 is practically a repeated sentence with the same citation

Author Response

Thank-you very much for taking the time to read this study and the proposed algorithm. Your feedback is very much appreciated and we have incorporated into the manuscript as follows: 

Line 344 vs 501
We have changed line 504 to the following: 
In deep learning, loss functions can enhance the performance of a model significantly, hence the evaluation of selecting a function which can advance the system is vital.
Introduction:
Thankyou for addressing the introduction, we have attempted to eliminate some text but we feel that the majority is important for background and motivational information. 

Thank you again for helping us improve our submitted manuscript.

Reviewer 2 Report

The manuscript presents a deep learning algorithm, called piNET, for the automated calculation of the Proliferation Index (PI) for Ki67, in the context of breast cancer diagnosis. This approach proved to be accurate and reliable across multi-center datasets. It also provided better results, on average, than other state-of-art methods.
The manuscript is well written and clear, the amount and quality of work are commendable, and the proposed approach can be useful for the scientific community. I recommend it for publication, with only minor changes.
Some little suggestions and comments:
Pag 2 Line 82: DAB, please define the acronym
Pag 6 Line 223: did you make a quantitative analysis to discriminate between ideal and non-ideal regions?
Pag 9 Line 331: “sigma = 3”. 3 what? 3 um? 3 pixels? Which is the unit of measurement?
Pag 10 Line 360: y(x1,x2). which quantity is y? It is a function of the coordinates of the nucleus center, but what kind of quantity is it?
Pag 13 Eq 8: The sum is over what? All the cohorts?
Pag 17 Line 571: “70:20:10”. Is it 70:10:20?
Pag 23 Fig 17: This figure is not mentioned in the text. Please add it.
Pag 24 Line 717: This sentence, with 2 verbs, does not make sense. Please rephrase it.
Pag 28 Line 849: GAN. Do you refer to Generative Adversarial Network? Please define the acronym.
Morever, despite the work is generally well written, there are many typos and grammatical errors. Here I will show you some of them, but I strongly recommend to carefully re-check the whole manuscript.
Pag 3 Line 100: due -> due to?
Pag 3 Line 129: tumour nuclei -> it is underlined
Pag 7 Line 245: in -> is?
Pag 9 Line 324: nuclei -> nucleus. As far as I know, the singular of nuclei is nucleus (Latin word), but you use nuclei across the whole manuscript. Please check it.
Pag 11 Line 383: two types of patch types -> simply two patch types?
Pag 13 Line 459: maybe a dot instead of the comma?
Pag 17 Line 553: are -> is
Pag 20 Line 628: the two “were” should be “was”
Pag 23 Line 696: used utilized -> repetition
Pag 23 Line 707: you are able to see it is able to -> please rephrase it
Pag 24 Line 719: “all most” -> I would remove “all”
Pag 24 Line 722: “this can be visually illustrated” -> as can be seen?
Pag 24 Line 729: repetition of accurately
Pag 24 Line 730: is ability -> is able
Pag 25 Line 738: “a large challenge being, variability” -> maybe without comma?

Author Response

Thank-you very much for taking the time to review the manuscript and for providing valuable feedback. The minor issues you had mentioned above have been analyzed and fixed. We have reviewed the manuscript in detail to ensure grammatical errors have been resolved.

Your time spent to review our manuscript and the feedback you have provided is very much appreciated! Thank-you again, for helping us improve our submitted paper.

Reviewer 3 Report

This is very interesting work to minimize the validity of evaluation for Ki67 that has been problem for the clinical use of Ki67. While utilizing of artificial intelligence is very promising in the field of pathology, to reach of the final diagnosis of malignancy is still difficult to overcome many issues. The PI by Ki67 is great field to utilize artificial intelligence .

Only one minor typo error is required to fix.

Line 824 

(DCIS)

Author Response

Thank-you very much for taking the time to review the manuscript and providing feedback. The minor typo has been fixed.